# Co-Crystallization Kinetics of 2:1 Benzoic Acid–Sodium Benzoate Co-Crystal: The Effect of Templating Molecules in a Solution

**Freshsya Zata Lini, Dhanang Edy Pratama and Tu Lee ***

Department of Chemical and Materials Engineering, National Central University, 300 Zhongda Road, Zhongli District, Taoyuan City 32001, Taiwan; fzatalini@gmail.com (F.Z.L.); dhanangedypratama09@gmail.com (D.E.P.)
* Correspondence: tulee@cc.ncu.edu.tw; Tel.: +886-3-4227151 (ext. 34204); Fax: +886-3-4252296

**Abstract:** The addition of dissolved templating molecules in crystallization will create "supramolecular assemblies" within the solution, serving as "anchor points" for the solute molecules to nucleate and grow. In this work, nucleation and crystal growth kinetics of 2:1 benzoic acid (HBz)–sodium benzoate (NaBz) co-crystallization with or without templates in a solution were analyzed by monitoring the concentration of the mother liquor during cooling crystallization. The results showed that the addition of the dissolved 2:1 or 1:1 HBz–NaBz co-crystals as templating molecules could reduce the critical free energy barrier of 2:1 HBz–NaBz co-crystal during its nucleation, but did not significantly affect the order of crystal growth rate. On the other hand, the critical free energy barrier of the nucleation process was increased if dissolved NaBz was used as a templating molecule, while a significant rise in the order of crystal growth rate occurred. The crystal habit obtained from the NaBz-templated system was needle-like, suggesting that sodium–sodium coordination chains of NaBz supramolecular assemblies in the solution phase were responsible for creating elongated crystals. Conversely, a large prismatic crystal habit found in non-templated and 2:1 and 1:1 HBz–NaBz co-crystal-templated systems implied that those templating molecules formed sparsely interconnected supramolecular assemblies in the solution phase.

**Keywords:** co-crystal; benzoic acid; sodium benzoate; nucleation; crystal; growth; kinetic; template; crystal habit

## 1. Introduction

Co-crystallization is one of the available techniques to alter the physicochemical properties of a molecule, where a molecule is paired with a suitable non-volatile molecule called co-former. These molecules are paired with each other in a stoichiometric ratio and form a crystal with distinct lattice parameters, differing it with solid solution. This pairing is facilitated by intermolecular hydrogen bonds, π–π interaction, halogen bonds, and Van der Waals forces, without any proton transfer, an important feature distinguishing it with salt formation reaction [1]. Co-crystallization is a feasible approach to tune the physicochemical properties of a substance, especially for a molecule that lacks easily ionizable functional groups or sensitive to acid and base treatment. Co-crystallization is an important technique in altering the materials properties of numerous active pharmaceutical ingredients, for example, acetaminophen–theophylline co-crystal [2], sulfathiazole–sulfanilamide co-crystal [3,4], sulfathiazole–theophylline co-crystal [3,4], nitrofurantoin–*p*-aminobenzoic acid co-crystal [5], ezetimibe–methyl paraben co-crystal [6], and benzoic acid–sodium benzoate co-crystal [7,8]. Co-crystallization is also gaining importance outside pharmaceutical field, with some of its applications in reducing the hygroscopicity of fertilizer [9] and improving the stability of an explosive agent [10].

Ensuring consistency is indispensable in large-scale co-crystallization process. The major implication of variabilities is the possibility of inconsistent properties of co-crystals

synthesized in separate batches. Taking caffeine–maleic acid co-crystals, for example, the solubility values of the stoichiometric ratios of 2:1 and 1:1 are starkly different [11]. In another study, a co-crystal with a single stoichiometric ratio was very difficult to obtain in a bench-scale stirred tank due to hydrodynamic mixing [12]. This evidence suggests that co-crystallization is highly process dependent.

To ensure a consistent production of co-crystals, understanding the kinetics of nucleation and growth of co-crystallization is necessary. In this study, nucleation and growth kinetics of co-crystals of benzoic acid (HBz) and sodium benzoate (NaBz) were studied. Their chemical structures are depicted in Figure 1. As drug substances, HBz is commonly used for acne treatment, antifungal agent, oral health care, and skin protectant, while NaBz is commonly found in menstrual or diuretic medication [13].

**Figure 1.** The molecular structure of (**a**) benzoic acid and (**b**) sodium benzoate.

Co-crystallization of HBz and NaBz was interesting to study because HBz and NaBz could form co-crystals in either a 2:1 [7,8,14] or a 1:1 stoichiometric ratio [15]. In addition, 2:1 HBz–NaBz co-crystal has two polymorphs: Form A and Form B [8]. The thermodynamic relationship between those two forms are enantiotropic with the A-to-B solid transition at about 110 °C. Form A is more stable at room temperature [8].

In crystallization, the presence of an additive may help the molecules to adopt a particular intermolecular interaction, promoting crystal nucleation and growth [16]. In extreme cases, additives could even induce a certain metastable form, which is hardly produced without the help of the additives. Sulfonamides have been utilized to dictate pyrazinamide to adopt a supramolecular synthon of sulfonamide–pyrazinamide linked via a N–H···O=C hydrogen bond, promoting the crystallization of $\gamma$-form pyrazinamide [17]. In another case, the various headgroups of the self-assembled monolayer of organic thiols deposited on a gold substrate could interact molecularly with sulfathiazole, resulting in the crystallization of various polymorphs [18]. Metacetamol, a molecule structurally similar to acetaminophen, was used as a templating molecule for producing metastable Form II of acetaminophen [19] by affecting the solution-mediated polymorphic transition. Cooling crystallization of acetaminophen in the presence of dissolved oxalic acid and maleic acid as templating molecules produced Form II of acetaminophen [20]. Dissolved templating molecules could form supramolecular assemblies with a packing pattern, more or less similar to the crystalline phase of the molecules [21]. We speculate that these supramolecular assemblies act as "invisible seeds" in a solution, influencing the outcome of a crystallization process.

The aim of this study is to use the templating molecules from dissolved co-crystals to guide the co-crystallization of the HBz–NaBz system, so that its variability of stoichiometric ratio and polymorphism may be overcome. Being an ionic co-crystal, 2:1 HBz–NaBz co-crystal is built on the coordination complexes of three kinds: a sodium atom with another sodium atom, a sodium atom with a carboxylic acid ligand, and a sodium atom with a carboxylate ligand and several hydrogen bonds, as illustrated in Figure 2. There are three hydrogen bonds located at $H_2 \cdots O_3$, $H_2 \cdots O_4$, $H_{6A} \cdots O_4$ in Form A, as illustrated in

Figure 2a [7]. For Form B, two hydrogen bonds are present: $H_2 \cdots O_3$ and $H_{5B} \cdots O_4$, as shown in Figure 2b [8]. To our best knowledge, no crystallographic data for 1:1 HBz–NaBz co-crystal have been published. Presumably, crystal packing also consists of sodium coordination complex(es) and hydrogen bond(s).

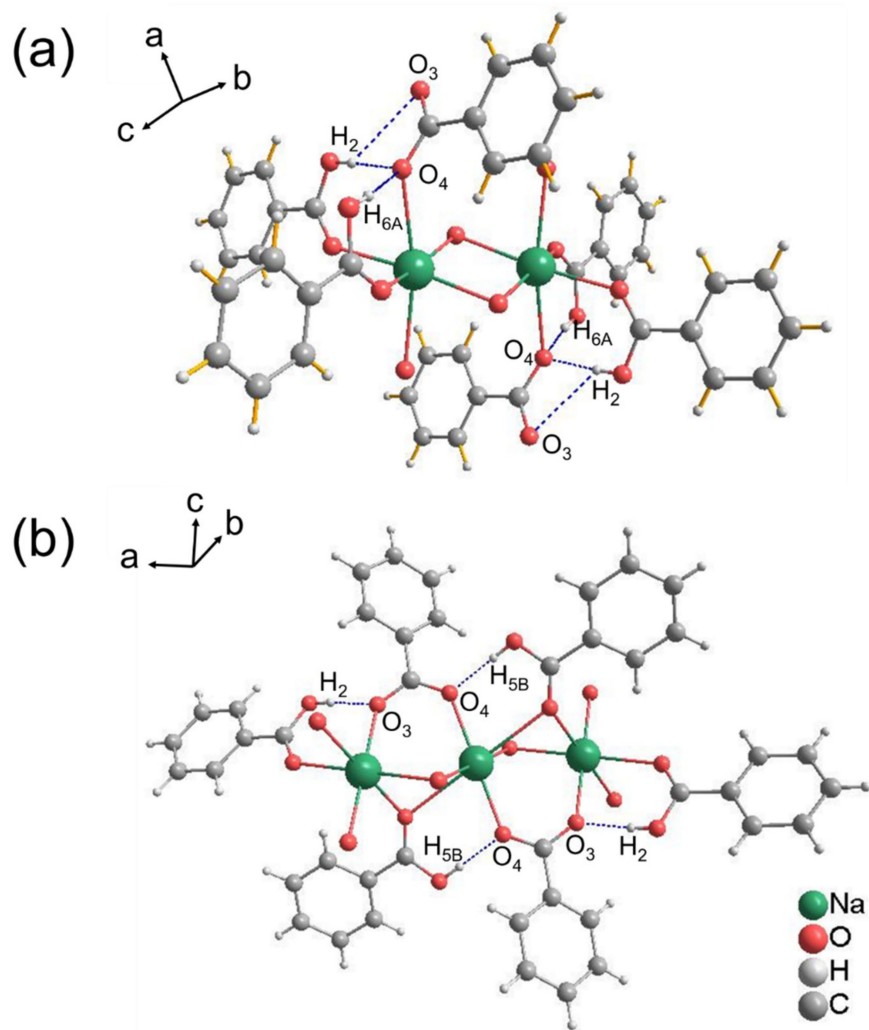

**Figure 2.** Illustration of intermolecular interactions of 2:1 co-crystal of HBz–NaBz: (**a**) Form I and (**b**) Form II. Some carboxylic/carboxylate ligands of sodium atoms are omitted for clarity.

In our work, the effects of supersaturation and templating molecules in solution towards the nucleation rate, crystal growth rate, stoichiometric ratio, and polymorphism of 2:1 HBz–NaBz co-crystals were studied. Simulating the intermolecular interactions between HBz and NaBz within the co-crystal lattice, templating molecules were originated from dissolved NaBz, 2:1 HBz–NaBz, and 1:1 HBz: NaBz co-crystals in solution. To ensure a homogeneous distribution of HBz and NaBz down to the molecular level, and to avoid pre-existed HBz or NaBz complexes in solution, reaction crystallization rather than re-crystallization of 2:1 HBz–NaBz co-crystal was employed. Therefore, the 2:1 HBz–NaBz co-crystals were produced by reacting three parts of HBz with one part of NaOH, resulting in HBz and NaBz in a 2:1 ratio, followed by cooling crystallization [22]. The scheme of this reaction co-crystallization is depicted in Scheme 1.

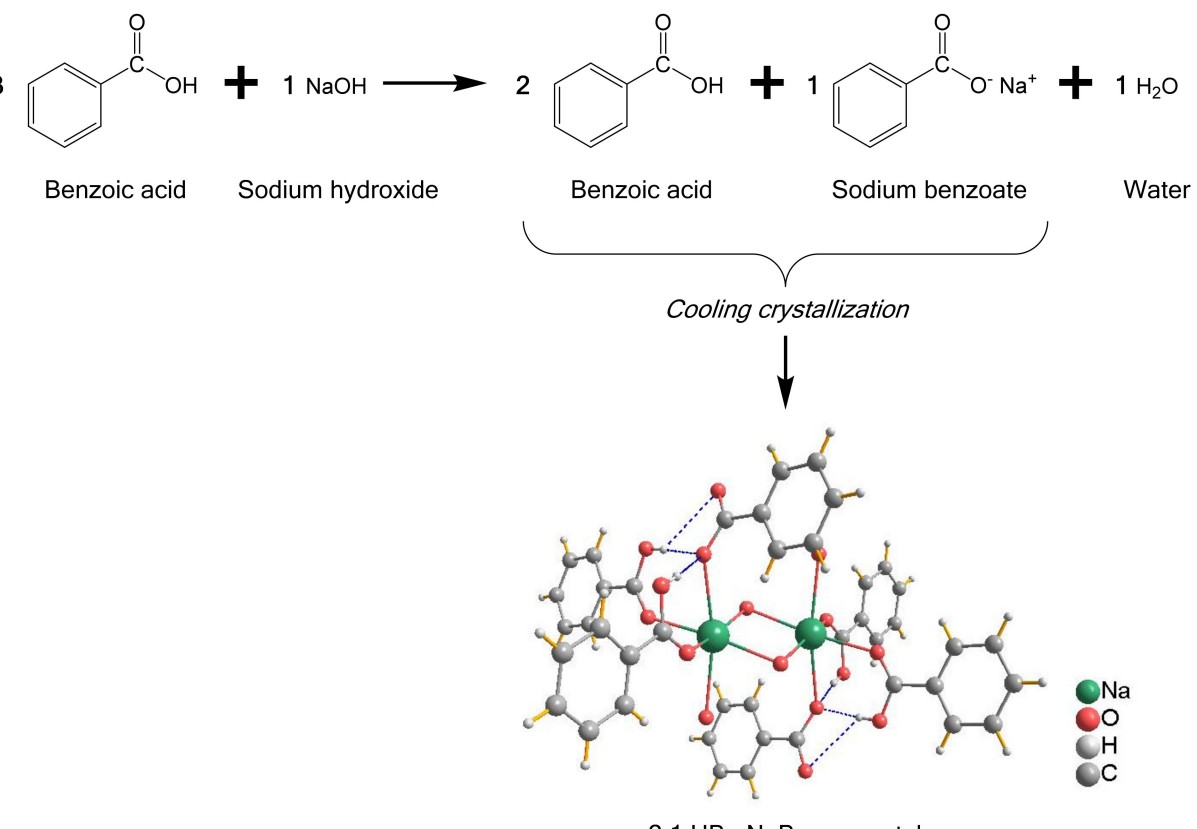

2:1 HBz-NaBz co-crystal

**Scheme 1.** Reaction co-crystallization of 2:1 HBz–NaBz co-crystal.

## 2. Theory

Nucleation rate: The time vs. concentration profile in a cooling crystallization process exhibited a Z-shaped curve, also known as a desupersaturation curve, as illustrated in Figure 3. An induction period is defined as the sum of the time required for a supersaturation to reach steady-state distribution of molecular clusters ($t_r$), to undergo primary nucleation ($t_n$), and to grow into nuclei with a detectable size ($t_g$) [23]. This relationship is given as:

$$\tau = t_r + t_n + t_g \tag{1}$$

Here, $t_r$ can be assumed to be zero because of rapid mixing. Primary nucleation time $t_n$ dominates over $t_g$ because of the relatively long plateau region compared with the shoulder portion near the turning point of the desupersaturation curve [24]. Therefore, the induction period, mainly consisting of primary nucleation can be assumed to be inversely proportional to the rate of primary nucleation per unit volume ($J$). This relationship can be expressed as:

$$\tau = t_n = f_N \cdot J^{-1} \tag{2}$$

where $f_N$ is the minimum detectable number density of nuclei per unit volume. For a detectable size of accumulated crystals of about ~10 µm, the $f_N$ value is about $7.64 \times 10^{11}$ nucleus m$^{-3}$ [25].

According to the classical nucleation theory (CNT), the overall free energy change in homogeneous nucleation, $\Delta G$, is the sum of the surface excess free energy, $\Delta G_S$, and the volume excess free energy, $\Delta G_V$ [23]. For a spherical nucleus, the relationship is given by:

$$\Delta G = \Delta G_S + \Delta G_V \tag{3}$$

$$\Delta G = 4\pi r^2 \gamma + \frac{4}{3}\pi r^3 \Delta G_v \tag{4}$$

where $r$ is the radius of the nucleus, $\gamma$ is the interfacial energy between the crystalline surface and the solution, and $\Delta G_v$ is the free energy change in the transformation per unit volume.

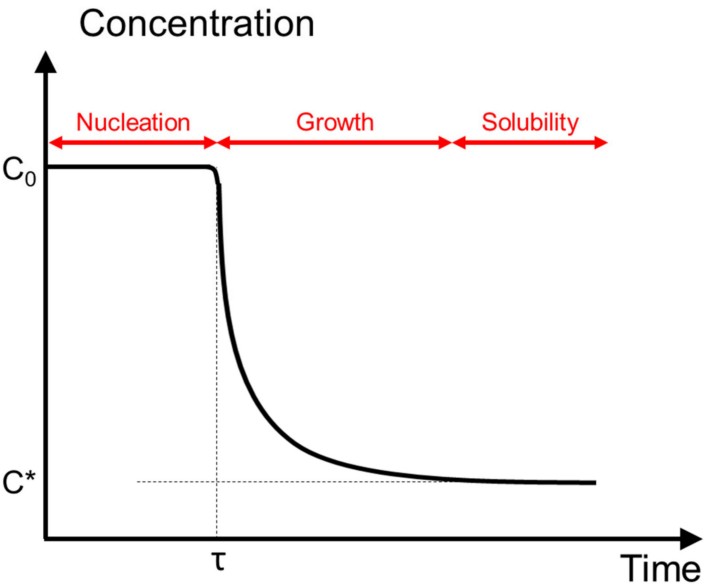

**Figure 3.** Desupersaturation curve of crystallization in general.

For the system to form a nucleus, it should pass the critical energy barrier ($\Delta G_{crit}$), which is equal to $d\Delta G/dr = 0$ for a spherical cluster [23]. Therefore, by applying this derivation to Equation (4), critical nucleus size ($r_{crit}$) can be obtained as:

$$r_{crit} = \frac{-2\gamma}{\Delta G_v} \tag{5}$$

By substituting Equation (5) into Equation (4), the free energy required to form a critical nucleus size ($\Delta G_{crit}$) can be obtained as:

$$\Delta G_{crit} = \frac{16\pi\gamma^3}{3(\Delta G_v)^2} = \frac{4\pi(r_{crit})^2}{3} \tag{6}$$

The growth of a non-electrolyte molecular cluster can be expressed by the Gibbs–Thomson relationship as [23]:

$$\ln \frac{C_0}{C^*} = \ln S_0 = \frac{2\gamma v}{kTr} \tag{7}$$

where $v$ is the molecular volume (molecular weight/(density $\times$ Avogadro's number)), $k$ is the Boltzmann's constant, and $T$ is the temperature. The density of 2:1 co-crystal of benzoic acid–sodium benzoate is 1387 kg·m$^{-3}$, as determined from the single crystal X-ray data of 2:1 co-crystal of benzoic acid–sodium benzoate [8].

According to Equation (5), $\Delta G_v$ is inversely proportional to the radius of a nucleus, $r$. By rearranging Equation (7) and substituting with $r$, $\Delta G_v$ can be rewritten as:

$$\Delta G_v = -\frac{2\gamma}{r} = -\frac{kT \ln S_0}{v} \tag{8}$$

By substituting the $\Delta G_v$ term in Equation (8) into Equation (6), the free energy required to form a nucleus in a critical size, $\Delta G_{crit}$, can be expressed as:

$$\Delta G_{crit} = \frac{16\pi\gamma^3 v^2}{3(kT \ln S_0)^2} \tag{9}$$

The free energy of forming a critical nucleus, $\Delta G_{crit}$, bears similarity with the temperature-dependent chemical reaction. Therefore, nucleation rate, $J$, can be written in the Arrhenius reaction rate equation as:

$$J = J_0\, exp\left(\frac{-\Delta G_{crit}}{kT}\right) \tag{10}$$

where $J_0$ is a pre-exponential factor. The rate of nucleation can be expressed alternatively by substituting Equation (9) into Equation (10):

$$J = J_0\, exp\left(\frac{-16\pi\gamma^3 v^2}{3(kT)^3(\ln S_0)^2}\right) \tag{11}$$

Since the induction period is inversely proportional to the rate of nucleation as described in Equation (2), taking the natural logarithm of Equation (11) and substituting it into Equation (2) would result in:

$$\ln \tau = \ln t_n = -\ln\frac{J_0}{f_N} + \left[\frac{16\pi\gamma^3 v^2}{3(kT)^3(\ln S_0)^2}\right] \tag{12}$$

The value of coefficient $J_0$ and $\gamma$ are calculated from the intercept and slope, respectively, of the linear regression of $\ln \tau$ vs. $(\ln S_0)^{-2}$. $\tau$ values used for the regression were obtained from the desupersaturation curve (Figure 3) at the designated $S_0$.

The theoretical number of molecules in the critical nucleus, $i^*$, is expressed as [26]:

$$i^* = \frac{4\pi r_{crit}{}^3}{3v} \tag{13}$$

Crystal growth rate: Once nuclei have become larger than the critical radius, they will be stable in the supersaturated solution. They will grow into visible crystals, while reducing the solute concentration in the mother liquor at the same time. This growth phenomenon would end once the concentration reaches the saturation point. The mass of the crystals, $m_t$, at a given time, $t$, can be written as:

$$m_t = (C_0 - C_t)\cdot V_{solution} \tag{14}$$

where $C_t$ is the concentration at a given time, $t$, and $V_{solution}$ is the volume of the solution.

There are two stages for incorporating the molecules into the crystal lattice: (1) diffusion of solute molecules from the bulk solution to the solid surface and (2) arrangement of the molecules into the crystal lattice. These two stages are difficult to measure individually. Hence, the crystal growth rate is commonly expressed using an "overall" crystal growth rate coefficient [23] as shown in Equation (15).

$$\frac{dm_t}{dt} = K_G A_t (C_t - C^*)^g \tag{15}$$

where $dm/dt$ is the crystal mass growth rate, $K_G$ is the overall crystal growth rate coefficient, $A_t$ is the crystal surface area at a given time $t$, $C_t$ is the concentration at any time $t$, $C^*$ is the saturation point of (i.e., solubility value) at 16 °C, and $g$ is the order of crystal growth kinetics. For 2:1 HBz–NaBz co-crystals, the value of $C^*$ in 4:1 (*v/v*) ethanol–water co-solvent at 16 °C was 0.196 kg/L.

Volume and surface area of a single crystal at a given time, $V_{t,i}$ and $A_{t,i}$, respectively, can be described by relating the characteristic length of the crystal $L_{t,i}$ with volumetric and surface shape factor $\alpha$ and $\beta$, respectively [27]. Both relationships are given in Equations (16) and (17):

$$V_{t,i} = \alpha \left( L_{t,i} \right)^3 \tag{16}$$

$$A_{t,i} = \beta \left( L_{t,i} \right)^2 \tag{17}$$

On the other hand, the volume of an individual crystal at a given time, $V_{t,i}$, can also be written in terms of its mass, $m_{t,i}$, and its density, $\rho_c$, as written in Equation (18).

$$V_{t,i} = \frac{m_{t,i}}{\rho_c} \tag{18}$$

Substituting Equation (18) into Equation (16) yields a relationship between the characteristic length $L_{t,i}$, mass $m_{t,i}$, and density $\rho_c$, as written in Equation (19). Substituting Equation (19) into Equation (17) resulted in Equation (20), relating term $A_{t,i}$ with $m_{t,i}$ and $\rho_c$.

$$L_{t,i} = \left( \frac{m_{t,i}}{\alpha \rho_c} \right)^{\frac{1}{3}} \tag{19}$$

$$A_{t,i} = \beta \left( \frac{m_{t,i}}{\alpha \rho_c} \right)^{\frac{2}{3}} \tag{20}$$

In the case of all crystals present in the system, the relationship becomes:

$$A_t = \sum A_{t,i} = \sum \beta \left( \frac{m_{t,i}}{\alpha \rho_c} \right)^{\frac{2}{3}} \cong \beta \left( \frac{m_t}{\alpha \rho_c} \right)^{\frac{2}{3}} \tag{21}$$

$A_t$ and $m_t$ denote the overall surface area of all crystals and total mass of all crystals in the system, respectively. By substituting Equation (21) into Equation (15), Equation (23) can be obtained after rearrangement and the introduction of the term $K_G{}'$. After taking its natural logarithm, Equation (24) can be rewritten as Equation (25).

$$\frac{dm_t}{dt} = K_G \left[ \beta \left( \frac{m_t}{\alpha \rho_c} \right)^{\frac{2}{3}} \right] (C_t - C^*)^g \tag{22}$$

$$\frac{1}{m_t^{\frac{2}{3}}} \frac{dm_t}{dt} = K_G \beta \left( \frac{1}{\alpha \rho_c} \right)^{\frac{2}{3}} (C_t - C^*)^g \tag{23}$$

$$\frac{1}{m_t^{\frac{2}{3}}} \frac{dm_t}{dt} = K_G{}'(C_t - C^*)^g \tag{24}$$

$$\ln \left( \frac{1}{m_t^{\frac{2}{3}}} \frac{dm_t}{dt} \right) = \ln K_G{}' + g \ln(C_t - C^*) \tag{25}$$

The value of coefficient $K_G{}'$ and crystal growth order $g$ are obtained by linear regression of $\ln \left( \frac{1}{m_t^{\frac{2}{3}}} \frac{dm_t}{dt} \right)$ vs. $\ln(C_t - C^*)$ using $C_t$ values within the growth period of desupersaturation curve (Figure 3) and $m_t$ values after calculation by Equation (14).

Nucleation and growth determination by concentration measurement: In studying the nucleation rate, the induction period is an important parameter to obtain. The induction period is commonly determined by visual confirmation of crystal appearance, such as eye [26], focused beam reflectance measurement [28], and turbidimeter [29–31]. Another method is by indirectly measuring the concentration change in the solution over time. Solute concentration can be measured by various means, such as density [32], refractive index [33], electrical conductivity [24], and ultrasound [34], resulting in the desupersatura-

tion curve as illustrated in Figure 3. The induction time would then be determined based on the plateau region. In this study, an off-line UV-Vis spectrometer was used as the main tool to measure the concentration of 2:1 HBz–NaBz co-crystal over time. The induction time would also be visually verified by off-line optical microscopy.

## 3. Results and Discussion

Nucleation kinetics: As predicted, all of the desupersaturation curves for co-crystalliz­ation with and without templating molecules exhibited a Z-shaped manner, as shown in Figure 4. All of the curves ended up as a constant flat line at the concentration of about 0.196 kg/L, which was the solubility value, $C^*$, of 2:1 HBz–NaBz co-crystals in the 4:1 ($v/v$) ethanol–water co-solvent at 16 °C. From these curves, $\tau$ was determined by measuring the time period of the plateau regions, as tabulated in Table 1. Both $S_0$ and $\tau$ values were converted into their natural logarithmic forms of $(\ln S_0)^{-2}$ and $\ln \tau$, respectively, yielding linear plots in Figure 5. According to Equation (12), values of $\gamma$ and $J_0$ can be obtained from the slopes and intercepts, respectively, as tabulated in Table 1. For calculating $J_0$, the $f_N$ value of $7.64 \times 10^{11}$ nucleus m$^{-3}$ was used [25]. This approximate $f_N$ value was considered to be acceptable since particles with sizes of about ~10 μm were detected upon reaching the induction period, based on the OM images in Figures S1–S4. Lastly, $\Delta G_v$, $\Delta G_{crit}$, $J$, $r_{crit}$, and $i^*$ were calculated by Equations (8), (9), (10), (5), and (13), respectively. The results are tabulated in Table 2.

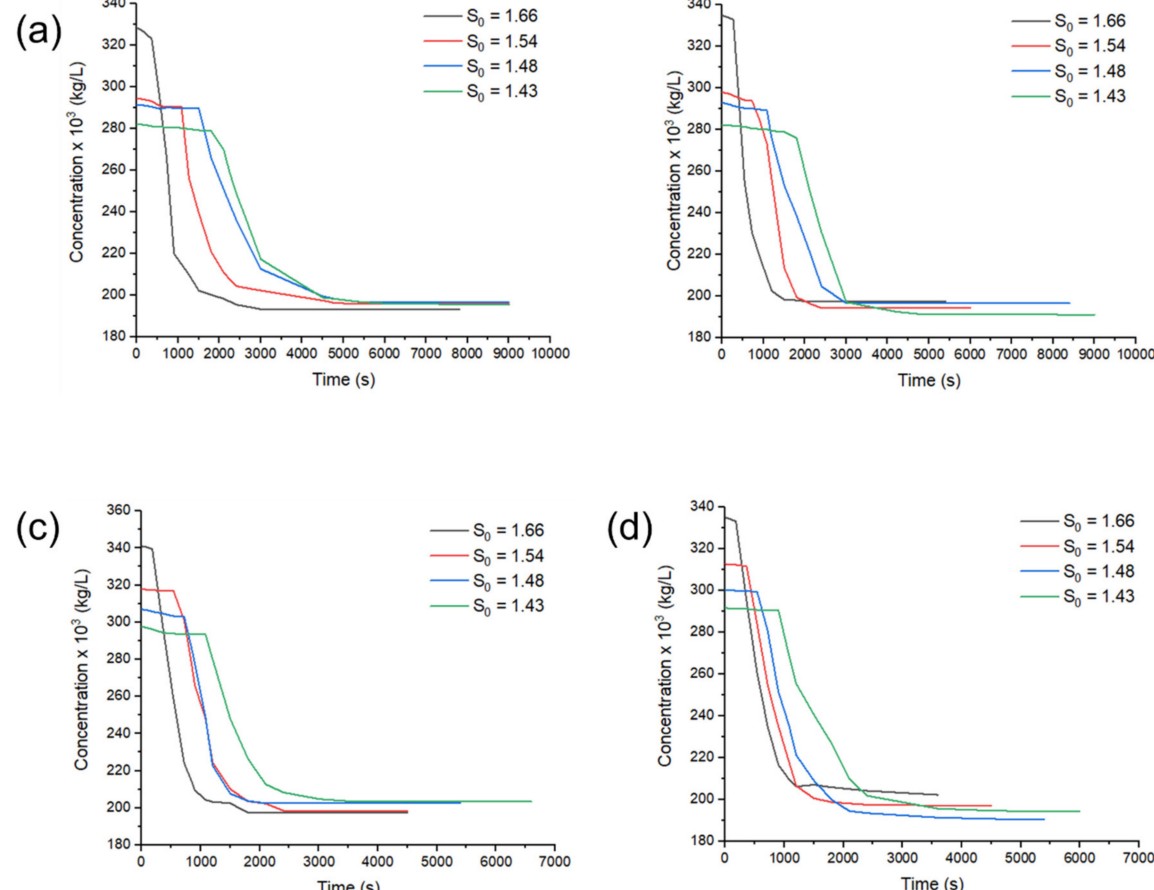

**Figure 4.** Representative desupersaturation curves of co-crystallization of 2:1 HBz–NaBz co-crystals at different $S_0$ values of 1.66, 1.54, 1.48, and 1.43 (**a**) without templating molecules addition, (**b**) with dissolved NaBz templating molecules addition, (**c**) with dissolved 2:1 HBz–NaBz co-crystal templating molecules addition, (**d**) with dissolved 1:1 HBz–NaBz co-crystal templating molecules addition.

**Table 1.** Induction period ($\tau$), interfacial energy ($\gamma$), and nucleation rate pre-exponential factor ($J_0$) of co-crystallization of 2:1 HBz–NaBz co-crystals at different initial degrees of supersaturation ($S_0$) with and without templating molecules.

| $S_0 = C_0/C^*$ | $\tau$ (s) | $\gamma \times 10^5$ (J·m$^{-2}$) | $J_0 \times 10^{-9}$ (Nucleus s$^{-1}$·m$^{-3}$) |
|---|---|---|---|
| **Without templating molecules** | | | |
| 1.66 | $370 \pm 14$ | | |
| 1.54 | $1080 \pm 27$ | | |
| 1.48 | $1494 \pm 49$ | $191.92 \pm 2.28$ | $8.04 \pm 0.30$ |
| 1.43 | $1848 \pm 180$ | | |
| **With NaBz templating molecules** | | | |
| 1.66 | $277 \pm 18$ | | |
| 1.54 | $714 \pm 47$ | | |
| 1.48 | $1086 \pm 39$ | $202.38 \pm 3.59$ | $15.97 \pm 2.79$ |
| 1.43 | $1816 \pm 68$ | | |
| **With 2:1 HBz–NaBz co-crystal templating molecules** | | | |
| 1.66 | $205 \pm 35$ | | |
| 1.54 | $538 \pm 38$ | | |
| 1.48 | $717 \pm 45$ | $194.34 \pm 3.37$ | $17.05 \pm 3.25$ |
| 1.43 | $1097 \pm 96$ | | |
| **With 1:1 HBz–NaBz co-crystal templating molecules** | | | |
| 1.66 | $187 \pm 20$ | | |
| 1.54 | $366 \pm 11$ | | |
| 1.48 | $533 \pm 32$ | $190.62 \pm 1.96$ | $18.62 \pm 2.26$ |
| 1.43 | $894 \pm 47$ | | |

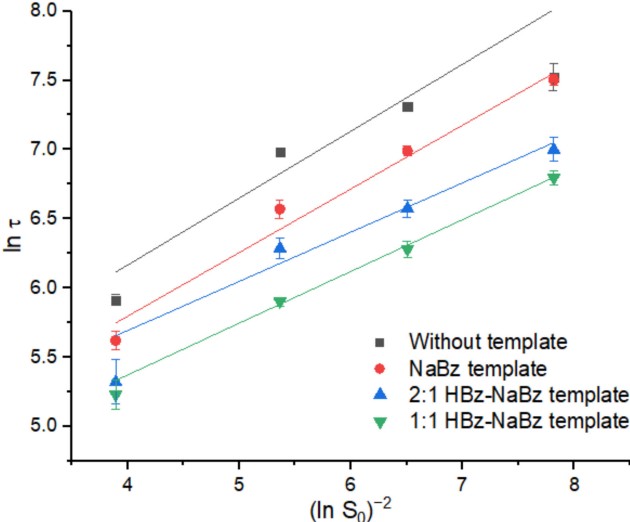

**Figure 5.** Plots of $(\ln S_0)^{-2}$ vs. $\ln \tau$ of co-crystallization of 2:1 HBz–NaBz co-crystal with and without templating molecules fitted to Equation (12) based on the data in Table 1.

As anticipated, supersaturation is inversely proportional to the induction period in both template and non-templated co-crystallization (Table 1). For a given $S_0$, the co-crystallizations with templating molecules had shorter induction periods than the ones without. Correspondingly, the nucleation rate, *J*, was faster for the templated co-crystallization. The descending order of the nucleation rate was co-crystallization with templating molecules of: (1) 1:1 HBz–NaBz co-crystal, (2) 2:1 HBz–NaBz co-crystal, (3) NaBz, and (4) without templating molecules. Since $\gamma$, $\Delta G_v$, $\Delta G_{crit}$, $r_{crit}$, and *i*\* parameters are independent from the induction period, the order of these parameters did not follow the nucleation rate order. At a given $S_0$, the co-crystallization system with NaBz templating molecules had the highest $\gamma$, $\Delta G_v$, $\Delta G_{crit}$, $r_{crit}$, and *i*\* values. The values of those parameters were almost the same for the co-crystallization system without templating molecules,

2:1 HBz–NaBz co-crystal templating molecules, and 1:1 HBz–NaBz co-crystal templating molecules.

**Table 2.** Free energy change in transformation per unit volume ($\Delta G_v$), free energy required to form a critical nucleus size ($\Delta G_{crit}$), nucleation rate ($J$), critical nucleus size ($r_{crit}$), and theoretical number of molecules in the critical nucleus ($i^*$) of co-crystallization of 2:1 HBz–NaBz co-crystals at different initial degrees of supersaturation ($S_0$) with and without templating molecules.

| $S_0 = C_0/C^*$ | $\Delta G_v \times 10^{-6}$ $(\text{J}\cdot\text{m}^{-3})$ | $\Delta G_{crit} \times 10^{22}$ $(\text{J})$ | $J \times 10^{-7}$ $(\text{Nucleus s}^{-1}\cdot\text{m}^{-3})$ | $r_{crit} \times 10^{11}$ $(\text{m})$ | $i^*$ |
|---|---|---|---|---|---|
| | | **Without templating molecules** | | | |
| 1.66 | −4.35 | 62.61 ± 2.22 | 167.34 ± 3.25 | 88.25 ± 1.05 | 6.19 ± 0.22 |
| 1.54 | −3.70 | 86.26 ± 3.06 | 92.56 ± 3.68 | 103.59 ± 1.23 | 10.01 ± 0.36 |
| 1.48 | −3.36 | 104.63 ± 3.71 | 58.45 ± 3.27 | 114.09 ± 1.35 | 13.38 ± 0.48 |
| 1.43 | −3.07 | 125.70 ± 4.46 | 34.51 ± 2.58 | 125.05 ± 1.48 | 17.62 ± 0.63 |
| | | **With NaBz templating molecules** | | | |
| 1.66 | −4.35 | 73.44 ± 3.89 | 250.35 ± 20.29 | 93.06 ± 1.65 | 7.26 ± 0.38 |
| 1.54 | −3.70 | 101.18 ± 5.36 | 124.62 ± 5.59 | 109.23 ± 1.94 | 11.75 ± 0.62 |
| 1.48 | −3.36 | 122.74 ± 6.50 | 72.54 ± 1.20 | 120.30 ± 2.13 | 15.69 ± 0.83 |
| 1.43 | −3.07 | 147.46 ± 7.81 | 39.04 ± 0.64 | 131.86 ± 2.34 | 20.66 ± 1.09 |
| | | **With 2:1 HBz–NaBz co-crystal templating molecules** | | | |
| 1.66 | −4.35 | 65.03 ± 3.36 | 330.00 ± 40.45 | 89.36 ± 1.55 | 6.43 ± 0.33 |
| 1.54 | −3.70 | 89.60 ± 4.63 | 177.76 ± 17.48 | 104.89 ± 1.82 | 10.40 ± 0.54 |
| 1.48 | −3.36 | 108.68 ± 5.62 | 110.00 ± 8.96 | 115.52 ± 2.00 | 13.90 ± 0.72 |
| 1.43 | −3.07 | 130.57 ± 6.75 | 63.48 ± 4.21 | 126.62 ± 2.20 | 18.30 ± 0.95 |
| | | **With 1:1 HBz–NaBz co-crystal templating molecules** | | | |
| 1.66 | −4.35 | 61.33 ± 1.88 | 398.61 ± 32.16 | 87.65 ± 0.90 | 6.07 ± 0.19 |
| 1.54 | −3.70 | 84.50 ± 2.59 | 222.79 ± 14.68 | 102.88 ± 1.06 | 9.81 ± 0.30 |
| 1.48 | −3.36 | 102.50 ± 3.14 | 141.80 ± 7.81 | 113.31 ± 1.17 | 13.10 ± 0.40 |
| 1.43 | −3.07 | 123.14 ± 3.78 | 84.48 ± 3.73 | 124.20 ± 1.28 | 17.26 ± 0.53 |

Undoubtedly, the nucleation rate of co-crystallization of 2:1 HBz–NaBz co-crystal was enhanced by templating molecules. The effect of these dissolved templates to the system were not just simply for raising the supersaturation, but these template molecules preserved their "memory" from the solid state in the form of supramolecular assemblies in the solution, in a similar fashion as rhenium compounds [21] and aspartic acid in the solution phase [35]. Those supramolecular assemblies aided co-crystallization by acting as anchoring sites for the immediate formation of hydrogen bonds and coordination complexes of a sodium atom with another sodium atom, a sodium atom with a carboxylic acid ligand, and a sodium atom with a carboxylate ligand among HBz and NaBz molecules, which were the backbone of 2:1 HBz–NaBz co-crystals. There are two pieces of evidence for the existence of supramolecular assembly in a solution. Firstly, simply redissolving 2:1 HBz–NaBz co-crystals at $S_0 = 1.66$ followed by cooling to 16 °C resulted in an instant precipitation of 2:1 HBz–NaBz co-crystals (i.e., $\tau < 60$ s), while reaction co-crystallization of HBz with NaOH, as previously described, gave a $\tau$ value of 370 s, which was six times slower. The rapid precipitation of 2:1 HBz–NaBz co-crystals after its dissolution was caused by the solid-state "memory effect" of the solutes, unlike the HBz–NaOH reaction co-crystallization solutes, which were freshly formed without any "memory effect" carried over to the solution. Secondly, yellow solution was produced right away upon mixing HBz–NaOH for reaction co-crystallization, whereas a normal clear solution was obtained by dissolving 2:1 HBz–NaBz co-crystals at $S_0 = 1.66$, as shown in the photograph in Figure S5. The appearance of the yellow solution strongly suggested that the supramolecular assembly derived from the reaction co-crystallization of HBz with NaOH was related to the π-stacking and charge transfer between the aromatic rings of HBz and NaBz in a solution. In general, the π-stacking arrangement and charge transfer are responsible for color change co-crystal systems of furosemide-4,4′-bipyridine [36] and emodin with various co-crystal formers [37]. In addition, a similar finding to our case of the yellow solution due to the π-stacking

of supramolecular assemblies has been documented in the acetaminophen-maleic acid co-crystal system [20].

In the case of employing the 1:1 HBz–NaBz co-crystal templating molecules, the nucleation rates were significantly faster than any other templating cases at a given $S_0$. In the HBz–NaBz co-crystal system, the requirement for sodium cation coordinations most likely could be satisfied by a 2:1 stoichiometric ratio [7]. Since the single-crystal structure of 1:1 HBz–NaBz co-crystal was unavailable, we speculate that the six-coordination bonds of sodium ions in 1:1 HBz–NaBz co-crystals were not as stable as the ones in the 2:1 ratio. To form a denser and a more stable crystalline structure, carboxylic group-containing HBz molecules would immediately be accepted to stabilize the coordination upon crystallization. The domino effect of rapid packing of the solutes to stabilize the template coordination had ultimately turned the clusters of the molecules into nuclei.

Interestingly, the values of $\gamma$, $\Delta G_{crit}$, $r_{crit}$, and $i^*$ of the 2:1 HBz–NaBz co-crystallization pointed out that NaBz templating molecules provided a slightly higher energy barrier in the nucleation process than without the templates. The sodium-rich NaBz supramolecular assemblies in the solution were mostly consisting of sodium–sodium and sodium–carboxylate coordination complexes, but without any sodium–carboxylic coordination or carboxylic–carboxylate hydrogen bonding moiety. It would take more efforts for both NaBz supramolecular assemblies and the solutes to rearrange their intermolecular interactions to accommodate both sodium–carboxylic coordination and carboxylic–carboxylate hydrogen bonding moiety, as reflected by the higher interfacial energy $\gamma$ and critical free energy barrier $\Delta G_{crit}$, as compared with those from non-templated and co-crystal molecules templated systems. The negligible differences of $\gamma$, $\Delta G_{crit}$, $r_{crit}$, and $i^*$ values in the cases of non-templated and templated co-crystallization using 2:1 and 1:1 HBz–NaBz co-crystals were because of the similar kinds of intermolecular interactions among the supramolecular assemblies and the solutes.

Growth kinetics: As plotted in Figure 4, all curves underwent a sharp plunge in concentration immediately after the end of the induction period. During this crystal growth period, the critical nuclei started to grow into visible crystals. The crystal growth rate, $dm_t/dt$, at a certain time $t$ was obtained by first converting the growth period of the desupersaturation curves in Figure 4 into the mass vs. time curves by Equation (14), fitting them with polynomials, and finally, taking the derivative of the polynomials with respect to time to obtain $dm_t/dt$. The parameters of crystal growth kinetics were obtained by plotting $\ln(m^{-2/3}(dm_t/dt))$ vs. $\ln(C_t - C^*)$, as in Figure 6, and linearly fitted them according to Equation (25).

The kinetics of crystal growth can be discerned in Table 3 by comparing the growth kinetic order, $g$, among different $S_0$s and various kinds of templating molecules. The $g$ values were relatively independent from $S_0$, with the average value of 1.2, 1.9, 1.0, and 1.0 for non-templated co-crystallization, templated co-crystallization by NaBz templating molecules, templated co-crystallization by 2:1 HBz–NaBz co-crystal templating molecules, and templated co-crystallization by 1:1 HBz–NaBz co-crystal templating molecules, respectively. Since the intermolecular interaction in the cases of non-templated and 2:1 and 1:1 HBz–NaBz co-crystal-templated systems were the same, the crystal growth would also happen in the same mechanism with each other, as reflected by the similarities in the $g$ values. On the other hand, the $g$ values in the NaBz-templated system was remarkably larger than other cases, meaning that the crystal growth rate happened faster than the other cases. We suspected that the excess coordination chains of sodium with another sodium atom in the NaBz-templated system contributed to the high growth rate order.

Characterizations of 2:1 HBz–NaBz co-crystals: All of the TGA scans of co-crystals synthesized at different $S_0$ values without templating molecules underwent weight losses between 90 to 250 °C, as illustrated in Figure S6, corresponding to the thermal decomposition of HBz [12]. Those weight losses of 59.6–63.1 wt.% were quite close to the theoretical weight loss of 62.9 wt.% (Equation (S1)). In the cases of co-crystallization aided with templating molecules, the TGA scans in Figure S7 also show that the 2:1 HBz–NaBz co-crystals

also exhibited weight losses of 61.9–63.1 wt.% between 90 and 250 °C, agreeing with the theoretical calculation of 62.9 wt.% based on Equation (S1).

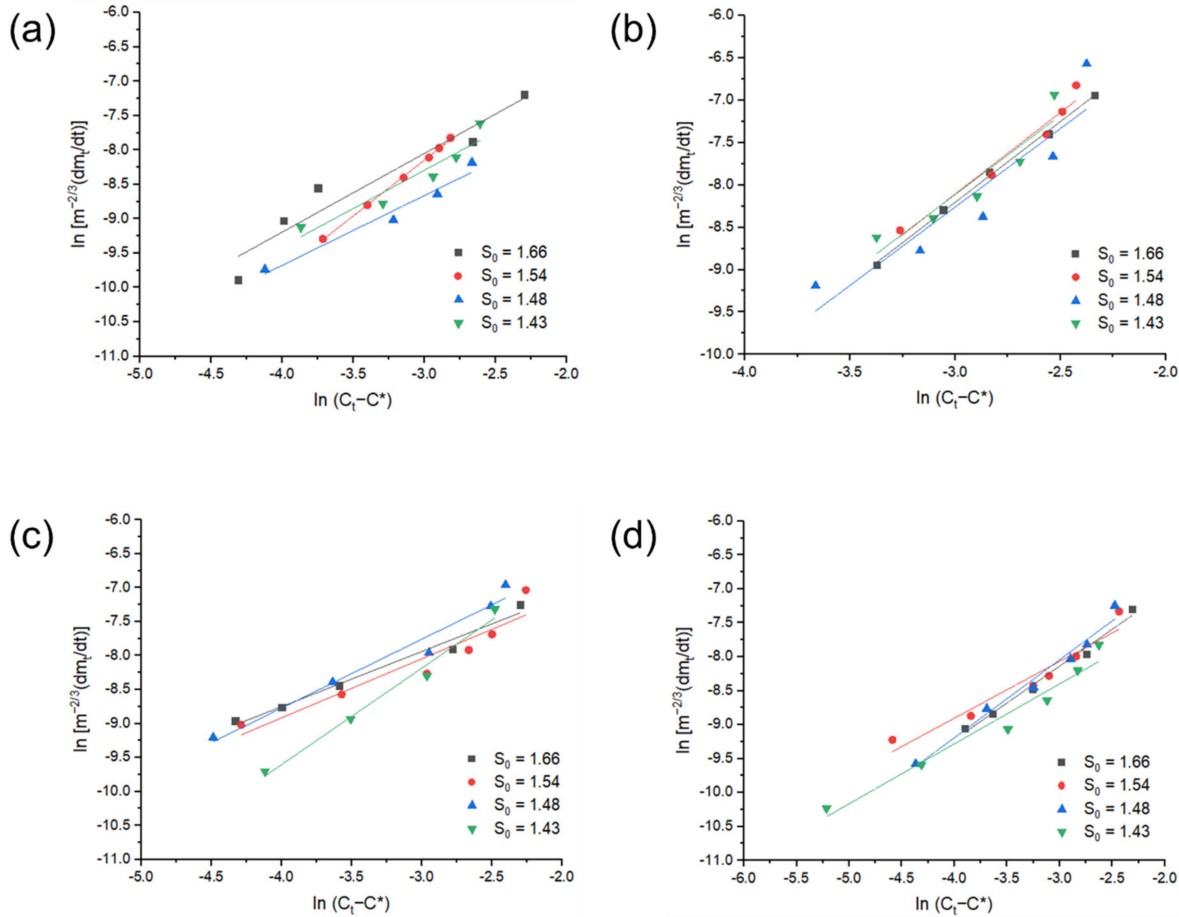

**Figure 6.** Plot of $\ln\left(m^{-2/3}(dm_t/dt)\right)$ vs. $\ln\left(C_t - C^*\right)$ of 2:1 HBz–NaBz co-crystallization by fitting to Equation (25): (**a**) without templating molecules, (**b**) with dissolved NaBz templating molecules, (**c**) with dissolved 2:1 HBz–NaBz co-crystal templating molecules, (**d**) with dissolved 1:1 HBz–NaBz co-crystal templating molecules.

The 2:1 HBz–NaBz co-crystals generated in all experiments without templating molecules were in the thermodynamically stable form of Form A, despite the variations in the initial degrees of supersaturation, $S_0$, and the templating molecules employed. The PXRD patterns of 2:1 HBz–NaBz co-crystals at different $S_0$ values in Figure S8 exhibited the diffraction characteristic peaks at $2\theta = 7.30°$, $7.99°$, $17.25°$, $18.70°$, $20.97°$, $24.52°$, and $27.42°$, matching well with the simulated PXRD pattern of 2:1 HBz–NaBz co-crystal Form A [8]. Similarly, the PXRD pattern in Figure S9 shows that Form A 2:1 HBz–NaBz co-crystals were produced at $S_0 = 1.66$ despite different templating molecules being used. From those results, it seemed that the addition of templating molecules in 2:1 HBz–NaBz co-crystallization could only enhance the rate of crystallization kinetics, without affecting either the stoichiometric ratio or the polymorphism of the 2:1 HBz–NaBz co-crystals.

**Table 3.** Order of crystal growth kinetics ($g$) and coefficient $K_G'$ of 2:1 HBz–NaBz co-crystallization at different initial degrees of supersaturation ($S_0$) with and without templating molecules.

| $S_0 = C_0/C^*$ | $g$ | $K_G' \times 10^3$ ($kg^{1/3}$ $s^{-1}$) |
|:---:|:---:|:---:|
| **Without templating molecules** | | |
| 1.66 | 1.1 | 9.9 |
| 1.54 | 1.6 | 38.0 |
| 1.48 | 1.0 | 3.6 |
| 1.43 | 1.1 | 7.0 |
| **With NaBz templating molecules** | | |
| 1.66 | 1.9 | 83.0 |
| 1.54 | 1.9 | 97.0 |
| 1.48 | 1.8 | 66.8 |
| 1.43 | 1.9 | 80.7 |
| **With 2:1 HBz–NaBz co-crystal templating molecules** | | |
| 1.66 | 0.8 | 4.1 |
| 1.54 | 0.9 | 4.3 |
| 1.48 | 1.0 | 8.9 |
| 1.43 | 1.4 | 19.7 |
| **With 1:1 HBz–NaBz co-crystal templating molecules** | | |
| 1.66 | 1.1 | 7.6 |
| 1.54 | 0.8 | 3.9 |
| 1.48 | 1.1 | 10.0 |
| 1.43 | 0.9 | 3.2 |

To further examine the templating effect on crystal habit, OM images of co-crystal samples at the end of the crystal growth phase, i.e., reaching the saturation concentration of 0.196 kg/L at 16 °C, were taken, and not later, to minimize the aging effect on crystal habit. The average aspect ratio of length-to-width of the 2:1 HBz–NaBz co-crystal was affected by the templating molecules. The representative OM images and the average aspect ratios of the crystals with and without templating molecules are shown in Figure 7 and Table 4, respectively. Normally, 2:1 HBz–NaBz co-crystals had a prismatic shape, such as the one in Figure 7a with the aspect ratio of about 6.0. Remarkably, the crystal habit was changed into almost a needle shape with NaBz templating molecules, with the aspect ratio of about 9.6 (Figure 7b). Other interesting findings were that the aspect ratios of the co-crystals were reduced to 5.3 and 4.3 with 2:1 and 1:1 HBz–NaBz co-crystal templating molecules, respectively.

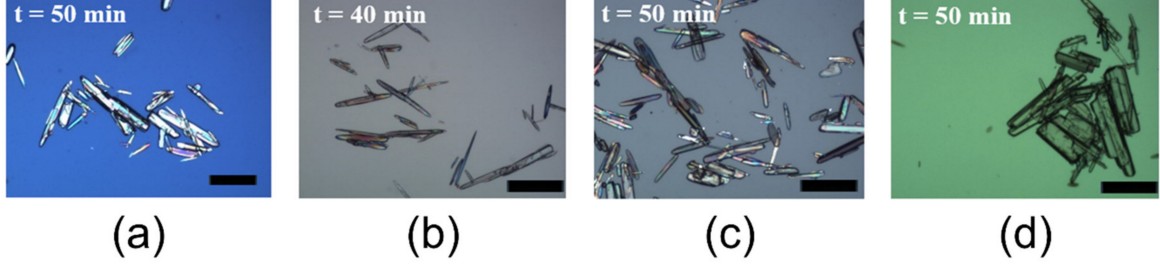

**Figure 7.** Representative OM images of 2:1 HBz–NaBz co-crystals generated at $S_0 = 1.66$: (**a**) without templating molecules, (**b**) with dissolved NaBz templating molecules, (**c**) with dissolved 2:1 HBz–NaBz co-crystal templating molecules, and (**d**) with dissolved 1:1 HBz–NaBz co-crystal templating molecules.

**Table 4.** Average aspect ratios of the 2:1 HBz–NaBz co-crystals generated with and without templating molecules at $S_0$ = 1.66.

| 2:1 HBz–NaBz Co-Crystals | Average Aspect Ratio |
|---|---|
| Without templating molecules | $6.0 \pm 1.7$ |
| With NaBz templating molecules | $9.6 \pm 3.7$ |
| With 2:1 HBz–NaBz co-crystal templating molecules | $5.3 \pm 1.4$ |
| With 1:1 HBz–NaBz co-crystal templating molecules | $4.3 \pm 1.5$ |

The theoretical crystal morphology of 2:1 HBz–NaBz co-crystals is depicted in Figure 8, determined based on Bravais–Friedel–Donnay–Harker (BFDH) theory. According to this theory, the most morphologically important faces are the one with the largest interplanar distance ($d_{hkl}$). For 2:1 HBz–NaBz co-crystals with monoclinic lattice and *2/m* symmetry [8], it was determined that those faces were (011), (002), (–102), (110), and (11–1). Superimposition of these faces onto the co-crystal packing configuration revealed that coordination complexes of sodium with another sodium atom formed elongated "chains" along [100] direction (Figure 8b), while several of these "chains" formed a discrete pattern along [010] and [001] directions (Figure 8c), weakly interconnected possibly by van der Waals forces.

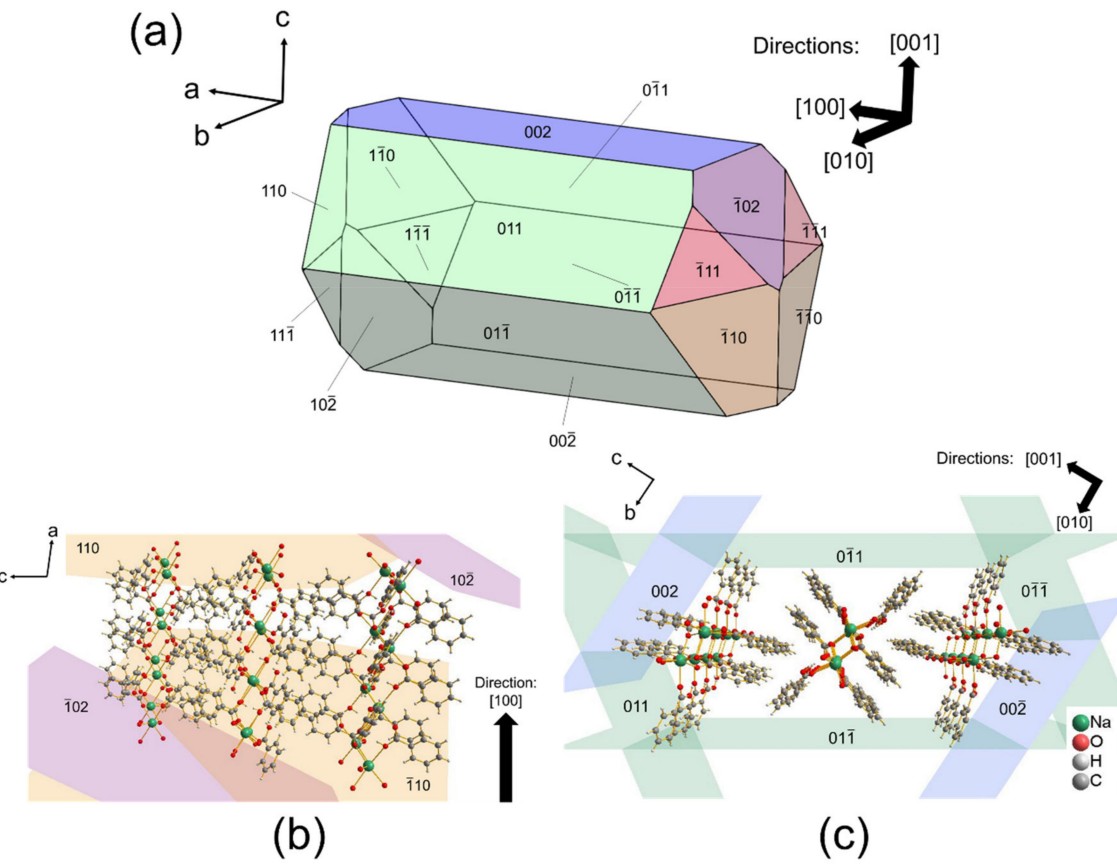

**Figure 8.** (**a**) Theoretical morphology of 2:1 HBz–NaBz co-crystal by BFDH method and theoretical molecular packing of 2:1 HBz–NaBz co-crystal in Form A, drawn together with the faces viewed along (**b**) *b*-axis and (**c**) *a*-axis.

During the co-crystallization with NaBz templating molecules consisting of elongated coordination complexes of sodium with another sodium atom or with a carboxylate ligand [38,39], these chains would serve as the attachment points for HBz and NaBz solutes to nucleate and grow. The generated co-crystals would then prefer to grow along [100] direction (Figure 8b), producing co-crystals with a high aspect ratio. It was also suspected

that these elongated coordination complexes of sodium "chains" were the reason behind the high growth order, *g*, of the NaBz-templated system (Table 3).

On the contrary, the 2:1 and 1:1 HBz–NaBz co-crystal templating molecules formed a higher quantity of discrete supramolecular assemblies, possibly interconnected with weak van der Waals forces. This resulted in the larger amount of "attachment points" for the solute molecules, resulting in crystals with larger size in the [010] and [001] directions (Figure 8c), making the aspect ratio to be smaller. This higher quantity of "attachment points" was also the reason behind the lower interfacial energy $\gamma$ for co-crystallization with both 2:1 and 1:1 HBz–NaBz co-crystal templating molecules.

## 4. Conclusions

Kinetics of co-crystallization of 2:1 HBz–NaBz co-crystal by cooling had been conducted by concentration measurements over time, producing Z-shaped desupersaturation curves as the basis in determining nucleation and crystal growth kinetics. The initial degree of supersaturation, $S_0$, was found to be proportional with the nucleation rate. The addition of 4.2 mol% of templating molecules enhanced the nucleation rate of 2:1 HBz–NaBz co-crystallization, regardless of the types of the templating molecules used. However, the addition of sodium-rich NaBz templating molecules could slightly increase the interfacial energy $\gamma$, critical free energy $\Delta G_{crit}$, critical nucleus radius $r_{crit}$, and the number of molecules in the critical nucleus $i^*$ due to the rearrangement of the coordination complexes of sodium with another sodium atom and carboxylate ligand within the supramolecular assemblies of the template to accommodate both sodium–carboxylic coordination and carboxylic–carboxylate hydrogen bonding moiety. On the other hand, the differences in those thermodynamic parameters among a non-templated system and 2:1 and 1:1 HBz–NaBz co-crystal templating molecules were negligible because the intermolecular interactions of the supramolecular assemblies of the template and the solute molecules were similar with each other.

Generally, the order of the crystal growth kinetics of 2:1 HBz–NaBz co-crystals depended on $S_0$ for the non-templated and NaBz-templated co-crystallization. The higher the supersaturation, the higher the order was. Meanwhile, the growth order was independent from $S_0$ for 2:1 and 1:1 HBz–NaBz co-crystal templating molecules. Analysis of the crystal habits and the aspect ratios of the generated 2:1 HBz–NaBz co-crystals revealed that the extensive coordination chain of sodium with another sodium atom within the NaBz supramolecular assemblies in the solution was responsible for creating elongated, almost needle-like crystals. In contrast, co-crystals with lower aspect ratios were generated in the presence of 2:1 and 1:1 HBz–NaBz co-crystal templating molecules. The lower aspect ratio was produced because the templating molecules formed sparsely interconnected supramolecular assemblies in the solution, creating many "anchor points" for the solutes, generating less-elongated crystals.

We believed that this work would spark the conversation about the existence of supramolecular assemblies in the solution phase. The history of the solution would greatly impact our understanding in various solution-based processes, such as mixing, crystallization, and dissolution. In the future, a decisive research work to directly observe, measure, and predict the "supramolecular assemblies" in various systems is much needed.

**Supplementary Materials:** These following Supplementary Materials are available online, free of charge at https://www.mdpi.com/article/10.3390/cryst11070812/s1. Materials and Methods, Figure S1: OM images of crystallization of 2:1 HBz–NaBz co-crystals without templating molecules at the initial degree of supersaturation ($S_0$) of: (a) 1.66, (b) 1.54, (c) 1.48, and (d) 1.43, Figure S2: OM images of crystallization of 2:1 HBz–NaBz co-crystals with NaBz templating molecules at the initial degree of supersaturation ($S_0$) of: (a) 1.66, (b) 1.54, (c) 1.48, and (d) 1.43, Figure S3: OM images of crystallization of 2:1 HBz–NaBz co-crystals with 2:1 HBz–NaBz co-crystal templating molecules at the initial degree of supersaturation ($S_0$) of: (a) 1.66, (b) 1.54, (c) 1.48, and (d) 1.43, Figure S4: OM images of crystallization of 2:1 HBz–NaBz co-crystals with 1:1 HBz–NaBz co-crystal templating molecules at the initial degree of supersaturation ($S_0$) of: (a) 1.66, (b) 1.54, (c) 1.48, and (d) 1.43, Figure S5: Photo

image of (a) clear solution obtained by dissolving 2:1 HBz-NaBz co-crystals in 4:1 (*v/v*) ethanol-water co-solvent, and (b) yellow solution derived by reaction co-crystallization of HBz and NaOH in 4:1 (*v/v*) ethanol-water co-solvent, Figure S6: TGA scans of 2:1 HBz-NaBz co-crystals generated without templating molecules at different initial degrees of supersaturation: (a) $S_0 = 1.66$; (b) $S_0 = 1.54$; (c) $S_0 = 1.48$; and (d) $S_0 = 1.43$, Figure S7: TGA scans of 2:1 HBz-NaBz co-crystals generated at $S_0 = 1.66$: (a) without templating molecules, (b) with dissolved NaBz templating molecules, (c) with dissolved 2:1 HBz-NaBz co-crystal templating molecules, (d) with dissolved 1:1 HBz-NaBz co-crystal templating molecules, Figure S8: PXRD patterns of 2:1 HBz-NaBz co-crystals generated without templating molecules from (a) $S_0 = 1.66$, (b) $S_0 = 1.54$, (c) $S_0 = 1.48$, and (d) $S_0 = 1.43$, and (e) simulated PXRD patterns of 2:1 HBz-NaBz Form A co-crystal (CCDC No. 875040), Figure S9: PXRD patterns of 2:1 HBz-NaBz co-crystals generated at $S_0 = 1.66$: (a) without templating molecules, (b) with dissolved NaBz templating molecules, (c) with dissolved 2:1 HBz-NaBz co-crystal templating molecules, and (d) with dissolved 1:1 HBz-NaBz co-crystal templating molecules, and (e) simulated PXRD pattern of 2:1 HBz-NaBz Form A co-crystal (CCDC No. 875040), Table S1: List of the theoretical amounts of produced 2:1 HBz-NaBz co-crystals and the amounts of HBz in Solution A and NaOH in Solution B required at different $S_0$ values and Table S2: Amounts of templating molecules added into Solution A for the co-crystallization of 2:1 HBz-NaBz co-crystals.

**Author Contributions:** Conceptualization, T.L.; methodology, F.Z.L. and T.L.; software, D.E.P.; validation, F.Z.L. and D.E.P.; formal analysis, F.Z.L. and D.E.P.; investigation, F.Z.L.; resources, T.L.; data curation, F.Z.L., D.E.P., and T.L.; writing—original draft preparation, F.Z.L.; writing—review and editing, D.E.P. and T.L.; visualization, D.E.P.; supervision, T.L.; project administration, T.L.; funding acquisition, T.L. All authors have read and agreed to the published version of the manuscript.

**Funding:** This work was supported by the Ministry of Science and Technology of Taiwan, R.O.C. under grant number of 104-2221-E-008-070-MY3, 107-2221-E-008-037-MY3, and 110-2221-E-008-006-MY3.

**Data Availability Statement:** Not applicable.

**Acknowledgments:** We are greatly indebted to the Precious Instruments Utilization Center of National Central University, especially to Li-Fan Chen for the assistance in TGA and Chin-Chuan Huang for the assistance in PXRD.

**Conflicts of Interest:** The authors declare no conflict of interest.

## List of Nomenclatures

| Symbol | Description | Unit |
| --- | --- | --- |
| $A_t$ | Overall crystal surface area at a given time | $m^2$ |
| $A_{t,i}$ | Surface area of an individual crystal at a given time | $m^2$ |
| $C_0$ | Initial concentration of 2:1 HBz–NaBz co-crystal | $kg\,L^{-1}$ |
| $C_t$ | Concentration of 2:1 HBz–NaBz co-crystal at a given time | $kg\,L^{-1}$ |
| $C^*$ | Solubility value of 2:1 HBz–NaBz co-crystal | $kg\,L^{-1}$ |
| $dm_t/dt$ | Crystal mass growth rate | $kg\,s^{-1}$ |
| $f_N$ | Minimum detectable number of nuclei per unit volume | $m^{-3}$ |
| $g$ | Order of crystal growth kinetics | dimensionless |
| $i^*$ | Theoretical number of molecules in the critical nucleus | dimensionless |
| $J$ | Rate of primary nucleation | $s^{-1}\,m^{-3}$ |
| $J_0$ | Pre-exponential factor of nucleation rate equation | $s^{-1}\,m^{-3}$ |
| $K_G$ | Overall crystal growth rate coefficient | $kg\,m^{-2}\,s^{-1}$ |
| $K_G{'}$ | Modified overall crystal growth rate coefficient | $kg^{1/3}\,s^{-1}$ |
| $k$ | Boltzmann's constant, $1.38065 \times 10^{-23}$ | $m^2\,kg\,s^{-2}\,K^{-1}$ |
| $L_{t,i}$ | Characteristic length of a single crystal at a given time | m |
| $m_t$ | Overall crystal mass | kg |
| $m_{t,i}$ | Mass of an individual crystal at a given time | kg |
| $r$ | Radius of a nucleus | m |
| $r_{crit}$ | Critical nucleus size | m |
| $S_0$ | Initial degree of supersaturation | dimensionless |
| $T$ | Temperature | K |
| $t_g$ | Time required for nuclei to grow into a detectable size | s |

| | | |
|---|---|---|
| $t_n$ | Time required for for the solutes to undergo primary nucleation | s |
| $t_r$ | Time required for a supersaturation to reach steady-state distribution of molecular clusters | s |
| $V_{solution}$ | Volume of solution | $m^3$ |
| $V_{t,i}$ | Volume of an individual crystal at a given time | $m^3$ |
| $v$ | Molecular volume | $m^3$ |
| $\alpha$ | Volumetric shape factor | dimensionless |
| $\beta$ | Surface shape factor | dimensionless |
| $\gamma$ | Interfacial energy between the surface and solution | $J\,m^{-2}$ |
| $\Delta G$ | Overall free energy change in homogeneous nucleation | J |
| $\Delta G_{crit}$ | Critical free energy required to form a critical nucleus size | J |
| $\Delta G_S$ | Excess free energy between the surface and bulk solid of a particle | J |
| $\Delta G_V$ | Excess free energy between bulk solid and solute in solution phase | J |
| $\Delta G_v$ | Free energy change in the transformation per unit volume | $J\,m^{-3}$ |
| $\rho_c$ | Density of crystal | $kg\,m^{-3}$ |
| $\tau$ | Induction period | s |

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
