# Peer review of "Co-Crystallization Kinetics of 2:1 Benzoic Acid–Sodium Benzoate Co-Crystal: The Effect of Templating Molecules in a Solution"

_crystals, doi:10.3390/cryst11070812_

Round 1
Reviewer 1 Report
See attached

Reviewer 2 Report
Please see the attached file.

Reviewer 3 Report
The Authors presented an approach on the nucleation and crystal growth kinetics od 2:1 benzoic acid and sodium benzoate co-crystallization with or without templates in a solution were analyzed by monitoring the concentration of the mother liquor cooling crystallization. The manuscript deserves to publish in Crystals after a minor correction. I would like to suggest introducing changes before publishing in Crystals.
The authors should revise in the manuscript as the following points:
- Abstract: The abstract should state briefly the purpose of the research, the principle results and major conclusions. The abstract should be corrected.
- Please review the article again. It has a lot of typos and punctuation errors.
- The article should be reorganized. All details of materials and equipment should be transferred to the supplementary materials.
- Figure 7: The authors presented a graph of the dependence of weight on temperature. Why is the weight in percentages?
- In the text, there are no reference to Figures 7 and 8 in the text.
- Please complete conclusions to combine theoretical work with future or proven experimental work.
